

# The Mesoamerican mid-summer drought: the impact of its definition on occurrences and recent changes

Edwin P. Maurer[1], Iris T. Stewart[2], Kenneth Joseph[3], Hugo G. Hidalgo[4]

[1]Civil, Environmental, and Sustainable Engineering Department, Santa Clara University, USA
[2]Department of Environmental Studies and Sciences, Santa Clara University, USA
[3]Department of Bioengineering, Santa Clara University, USA
[4]Universidad de Costa Rica, Centro de Investigaciones Geofísicas (CIGEFI) y Escuela de Física, San José, Costa Rica

*Correspondence to*: Edwin P. Maurer (emaurer@scu.edu)

**Abstract.** The mid-summer drought, veranillo or canícula, is a phenomenon experienced in many areas, including Mexico, Central America, and the Caribbean. It generally is experienced as reduced rainfall in July-August, in the middle of the typical rainy season (May–September). Many past studies have attempted to quantify changes in mid-summer drought

characteristics during the recent past or for future climate projections. To do this, objective definitions of a mid-summer drought's occurrence, strength and duration have been developed by many researchers. In this effort we adopt a recent set of definitions and examine the impact of varying these on the characterization of mid-summer droughts and the detected changes over the past four decades. We find the selection of a minimum intensity threshold has a dramatic effect on the results of both the area considered as experiencing a mid-summer drought and the changes detected in the recent historical

record. The intensity chosen can affect both the magnitude and direction of changes reported in the recent observed record. Further, we find that the typical mid-summer drought pattern may not be occurring during the time it has historically; whether examining past or future changes or developing improved seasonal forecasts, the non-stationarity of its timing should be accommodated.

## 1 Introduction

In many parts of Mexico and Central America (usually on the Pacific slope) there is a well-defined summer rainy season, often marked by an early and late peak period separated by a brief period of reduced rainfall. This reduced rainfall event, which typically persists for 2-4 weeks in July-August, is often referred to as the mid-summer drought (MSD) in the climate science community. In Central America it is referred to by locally distinct names, such as the *veranillo* or *canicula* (Magaña et al., 1999; Maldonado et al., 2016). Variability in different characteristics of the MSD is well established (García-Oliva and

Pazos, 2021) and can have important agricultural and economic consequences for the region, especially in the area denoted as the Central American Dry Corridor (Hidalgo et al., 2019; Stewart et al., 2021).



Many MSD descriptions have originated with the smallholder farmers in the region and often have qualitative components. While in specific locations an MSD definition may be defined historically using specific dates, such as July 15-August 15, 35 the regional variability in those dates and their inflexibility for representing change in MSD timing make their use in studies such as ours impractical (Alfaro, 2002; Curtis, 2004; Magaña et al., 1999). In many regions of Central America, the timing and magnitude of the early and late rainy periods are critical for a first and possible second planting season; subsistence farmers who mostly rely on rain-fed agricultural practices must time their planting and harvesting to anticipate the end of the MSD and the arrival of a second peak of rainfall. How the presence of an MSD pattern, its timing, intensity and duration are 40 affected by climate variability and change therefore is intimately tied to the agricultural cycle and farmer livelihoods.

Because of this regional importance, there have been many studies of the MSD, both examining the recent record to detect trends in its characteristics (e.g., Anderson et al., 2019), and looking toward the future to discern what a disrupted climate might produce (e.g., Corrales-Suastegui et al., 2020; Maurer et al., 2017; Rauscher et al., 2008; Vichot-Llano et al., 2021). 45 When considering either current or future MSD characteristics and metrics to evaluate these, most studies adopt at least some of the methods established by Karnauskas et al. (2013) using monthly gridded data, or Alfaro (2014) using daily station data, including the timing, intensity, and duration of the MSD. However, the details in the definitions of what constitutes an MSD pattern and the quantification of MSD characteristics are less consistently defined. Some measures of past changes, as well as future projections, can be significantly affected by subtle changes in definitions of the MSD. For 50 example, the definition of the timing when rainfall minima and maxima need to occur will affect whether a given year or location is counted as experiencing an MSD. In addition, temporal and spatial scales play a role in determining the existence of an MSD pattern. Zhao and Zhang (2021) found the existence of an MSD signal in some locations in Central America and Mexico dependent on whether a method used daily or monthly data.

An understanding of where in the study region an MSD pattern exists, and how it has been impacted by recent climate variability and change, has been elusive - at least partly due to limited mathematical descriptions of the phenomenon, and by the lack of an exploration of the effects of variation of the parameters used for the determination of whether an MSD phenomenon is present. In addition, any assumption of how frequently an MSD pattern must be identified to declare a given area as being dominated by MSD is arbitrary, yet will impact the area considered to be MSD as well as the area where 60 climate change might have affected the presence of characteristics of the MSD. Therefore, the impact of the mathematical definition as well as that of climatic change on MSD extent must be explored jointly.

A recent study (Anderson et al., 2019) used pentadal precipitation data from the quasi-global CHIRPS data set, covering Guatemala, Honduras, Nicaragua, and El Salvador. For 1981-2018 they found significant trends in the duration of the MSD 65 in many locations, but most other MSD characteristics did not show discernible trends. As Anderson et al. (2019) note, there may be a disconnect between statistically significant changes in objectively-defined MSD conditions and the experience and

understanding of the phenomenon by smallholder farmers, especially in the northern part of their domain in Guatemala and Mexico. The importance of extending a study domain of the MSD into more of Mexico is supported by recent studies characterizing its influence in the historical record (Perdigón-Morales et al., 2018) and potential changes in a disrupted

climate in the northern, water-limited, and primarily agricultural regions of Mesoamerica (Corrales-Suastegui et al., 2020; Stewart et al., 2021). However, we are not aware of a study that has examined the sensitivity of the MSD spatial and temporal extent to its definition, and the impact the definition has on the presence of Central America wide changes during the most recent warming.

In this effort, we build on the past work to improve an objective, mathematical definition of the MSD that includes measures to evaluate the magnitude and timing of the phenomenon and to characterize the variability, trends, and changes in the spatial domain with an MSD pattern during the recent historical record. In particular we: 1) Use an expanded domain, as compared to previous studies, that includes Central America and Mexico, and potentially exhibits MSD characteristics; 2) Use daily data rather than monthly or pentadal aggregated data to characterize the MSD with finer precision; 3) Build on past

work to refine definitions of MSD characteristics and spatial extent; and 4) Explore the effect of parameter variability in the MSD definition on the magnitude, direction, and changes during the recent observational record (1981-2020), which includes the warmest years in the observational record. Our work is motivated by the need for better understanding of past changes that align with smallholder experience, for seasonal forecasts of specific MSD features, and for projections on how the MSD may change through the 21st century.

**2 Methods and Data**

We use precipitation-based definitions of the MSD, consistent with many past studies (Alfaro, 2014; Anderson et al., 2019; Karnauskas et al., 2013). The primary data source we use is the gridded daily precipitation product of the Climate Hazards group Infrared Precipitation with Stations (CHIRPS) v.2.0 dataset (Funk et al., 2015), aggregated to 0.25° (approximately 25 km) as described by Stewart et al. (2021). CHIRPS is developed by the Climate Hazards Group at the University of

California, Santa Barbara and the U.S. Geological Survey Earth Resources Observation and Science Center. Daily, monthly, and seasonal products are built around blending satellite Cold Cloud Duration observations and improved interpolation techniques of high resolution, long period-of-record precipitation estimates. CHIRPS forms the basis for the U.S. Agency for International Development's Famine Early Warning Systems Network. The limited use in this study of temperature data uses the related CHIRTS dataset (Funk et al., 2019).


In some recent studies, the inclusion of temperature in the analysis of the MSD has been recognized as important due to the vulnerability of the affected areas to soil moisture (Romero et al., 2020), reflecting the water deficit and warmer temperatures experienced by farmers. This was a motivation in one study for the use of a 'hydrologic satisfaction' threshold

(MAGFOR, 2010) to define the intensity of an MSD episode. For changes in the recent observed record in the study region,
however, the influence of temperature variability on changes in drought indices is much smaller than that of precipitation changes (Stewart et al., 2021). For this reason, we only consider precipitation-based definitions for the MSD, though projections of future changes, when temperature changes will become more pronounced, should consider alternate definitions that include accounting for temperature increases.

To define whether an MSD occurs in any year and quantify its important features, we started with the method of Anderson et al. (2019) and modified it to work with our daily data set. For each calendar year of daily precipitation data we follow these steps: 1) smooth the data using two passes of a 31-day triangular filter; 2) locate the minimum (which must be an inflection point) between June 1st and August 31st (window 1); 3) check that the minimum from step 2 is also the minimum between May 1 and October 31 (window 2); 4) locate the highest peak between January 1 and minimum date; 5) locate the highest
peak between the minimum date and December 31; 6) if the two peaks from steps 4 and 5 are not within the May 1st to October 31st period the year is not an MSD; 7) if those two peaks are not separated by a defined minimum duration (e.g., 15 days) the year is not an MSD; 8) if the average of the maxima minus minimum is not greater than a defined minimum intensity (e.g., 3 mm), the year is not classified as an MSD. The order in which these constraints are applied to the data and the magnitude of the parameter values matters. Figure 1 presents a flowchart illustrating these steps.


Finally, to define whether a location is classified as having an MSD, a threshold is defined for the percentage of years with an MSD according to the definition above. Anderson et al. (2019) set this at 33 out of 38 years (87%), since they used a 38-year precipitation record (1981-2018). This study uses as a baseline that 80% of the years must exhibit an MSD for it to be classified as an MSD cell, and we explore the effect of changing this value.


To compare how and where the MSD pattern has changed over the recent past, we divide the study period into two equal 20-year periods, namely (1981 - 2000) and (2001-2020) and compare average conditions in precipitation and MSD presence between the earlier and the later period.





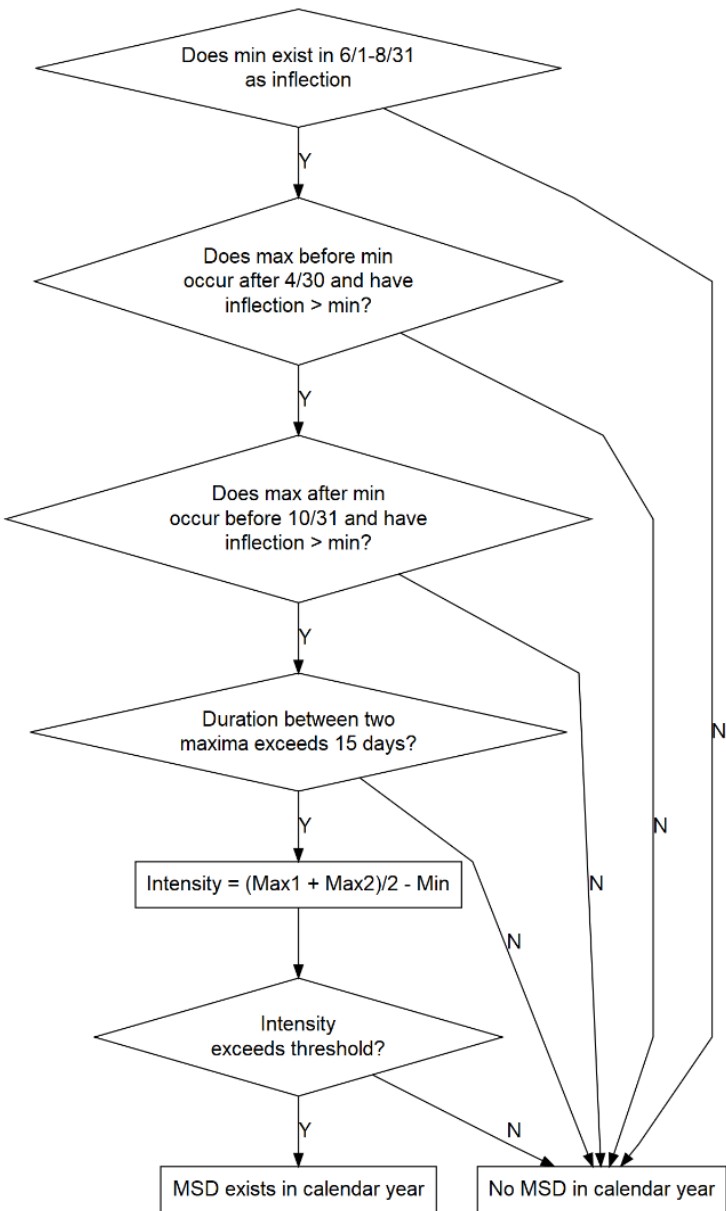

**Figure 1: The steps used to determine for each calendar year whether a location experiences an MSD.**

Our "original" values for these characteristics are summarized in Table 1. These are designed to reproduce as closely as possible the methods of Anderson et al. (2019). These values are later varied to explore their influence on the extent of cells characterized as having an MSD and the impact on detected changes between 1981-2000 and 2001-2020.



**Table 1: Original values for key components of the MSD definition.**

| | |
|---|---|
| Minimum duration | 15 days |
| Minimum intensity | 3 mm/day |
| Window 1 | June 1st - August 31 |
| Window 2 | May 1st - October 31 |
| % of years | 80% (32 of 40 years or 16 of 20 years) |

These original values are adjusted to assess the influence of specific definitions on the determination of whether an MSD exists in a location, and whether statistically significant changes have occurred over the recent historical record.

Statistical tests consist of comparing the proportions of MSD years in a 20-year period using Fisher's exact test (Mehta and Patel, 1983), and comparing the central tendency of statistics between two 20-year groups using a Wilcoxon (Mann-Whitney) signed-rank test (Helsel et al., 2020). Statistical significance is evaluated at a 5% level ($\alpha = 0.05$).

**3 Study Area**

The Mesoamerican region of Central America and Mexico is a region with very distinct, but spatially highly variable climatic patterns. Figure 2a shows the climatological precipitation pattern across the study domain. The CHIRPS data in this figure have been aggregated spatially and temporally (to monthly averages). Figure 2a illustrates the tremendous variability in climate that exists in the region, from wet tropical climates to cold arid regions in Mexico's highlands. Despite these differences, many regions are characterized by a highly seasonal climate, with a pronounced spring dry season followed by a summer (June - September) rainy season with monthly precipitation of 400-500 mm or more. A clear dip in July-August precipitation associated with an MSD pattern is apparent in many of the grid boxes of the domain.

There are parts of Mexico and of the Caribbean, regions excluded from many prior MSD studies, which have exhibited the canonical MSD pattern in the past (Perdigón-Morales et al., 2018). While warmer temperatures during the summer months prevail for the northern parts of the study area, the seasonal cycle of temperature is muted for areas closer to the equator and/or to the coast, as would be expected, and with a few exceptions for high-elevation regions in the domain, temperatures remain well above freezing throughout the year. In Fig. 2b the precipitation changes between the early (1981-2000) and late record (2001-2020) vary widely across the domain. Particularly in Honduras, Nicaragua and Costa Rica a decline is evident in precipitation in the wet season, including the July-August period of the MSD. Warming on the order of 1-2 °C has





generally taken place throughout the domain, but temperature changes are highly variable month-to-month; the changes are broadly positive and statistically significant (Fig 2b; Stewart et al., 2021).

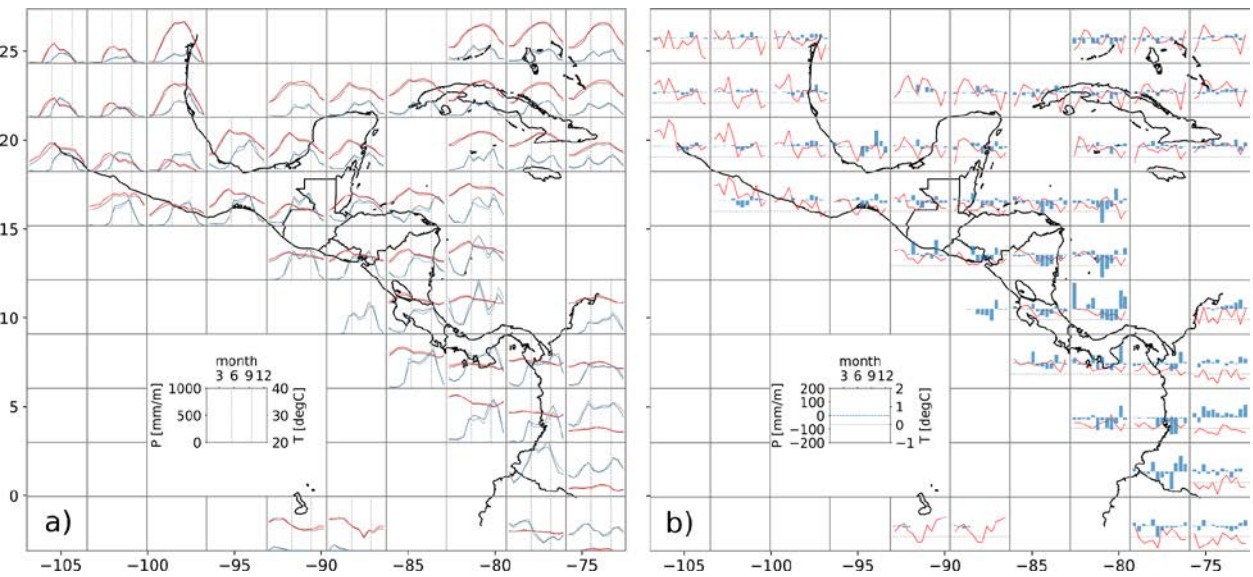

**Figure 2: The study area with a) climatological patterns of monthly precipitation (blue) and temperature (red) for the 1981 - 2020 period, and b) changes between the early (1981-2000) and later (2001-2020) part of the study periods, based on the CHIRPS (precipitation) and CHIRTS (temperature) data sets.**

## 4 Results

Applying our original definition of the MSD as defined in Table 1 and Fig. 1 yields Fig. 3.





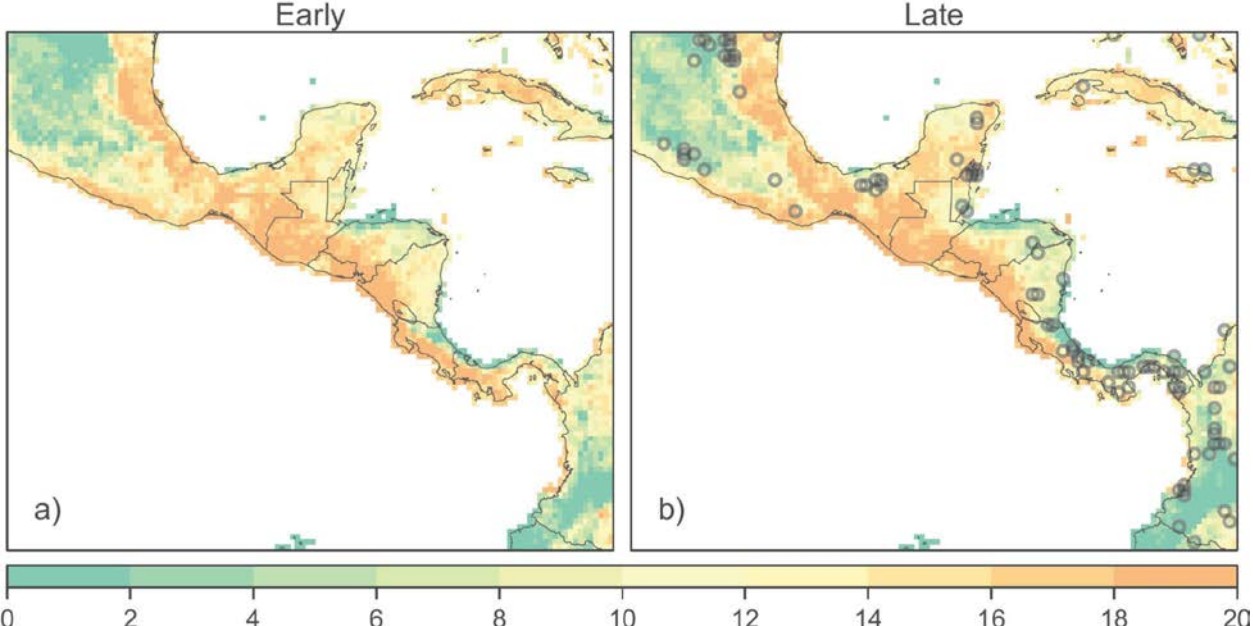

**Figure 3: Number of years out of 20 in a) the early period (1981-2000) and b) the late period (2001-2020). Circles in panel b indicate the difference in proportions of MSD years in the two periods is statistically significant based on Fisher's exact test.**

In Fig. 3, two-thirds of the grid cells with statistically significant changes are locations that are dominated by an MSD in the early period but not in the late period, based on the original values of the MSD definition in Table 1. Aside from some areas on the Caribbean side of Mexico where MSD years have become more frequent, most of the significant changes are concentrated in the southern part of the domain (especially Panama), suggesting a change in precipitation seasonality in southern Central America. This is explored further below.

Summing MSD presence over the early and late period of study results in Fig. 4, which (by design) closely resembles that of Anderson et al. (2019). Figure 4 shows most of the pixels that are classified as experiencing an MSD pattern exhibit this throughout the past four decades. Expansion of the area with an MSD occurs in the northern part of the domain, mostly in Mexico's Yucatan Peninsula. Areas with MSD in the early period but not in the later period appear generally toward the southern part of the domain, especially evident in Panama. More intense drying of the Northern part of Central America, with intensification of the MSD, has been identified as potentially indicative of a southern shift in the summer location of the ITCZ (Rauscher et al., 2011; Hidalgo et al., 2013), also an anticipated impact of climate disruption on this region (Rauscher et al., 2008). The regions classified as MSD in the late but not the early period also coincide largely with the areas showing significant trends for greater MSD intensity (Fig. A1), due in part to an increase in the magnitude of the precipitation peaks, especially the first peak (Figs. A2 and A3), and a decline in the intervening minimum (Fig. A4).



While Mexico and Central America are often the focus of MSD studies, our analysis using consistent criteria demonstrates
that the phenomenon is also widely present in the Caribbean, as shown by others (e.g., Almazroui et al., 2021), though the
driving mechanisms for the MSD are distinct from the rest of the domain (Curtis and Gamble, 2008). For the portions of the
Caribbean included in Fig. 4 there are few areas showing a tendency for increased MSD occurrence, though many do see a
continuing MSD classification for both the early and late periods.

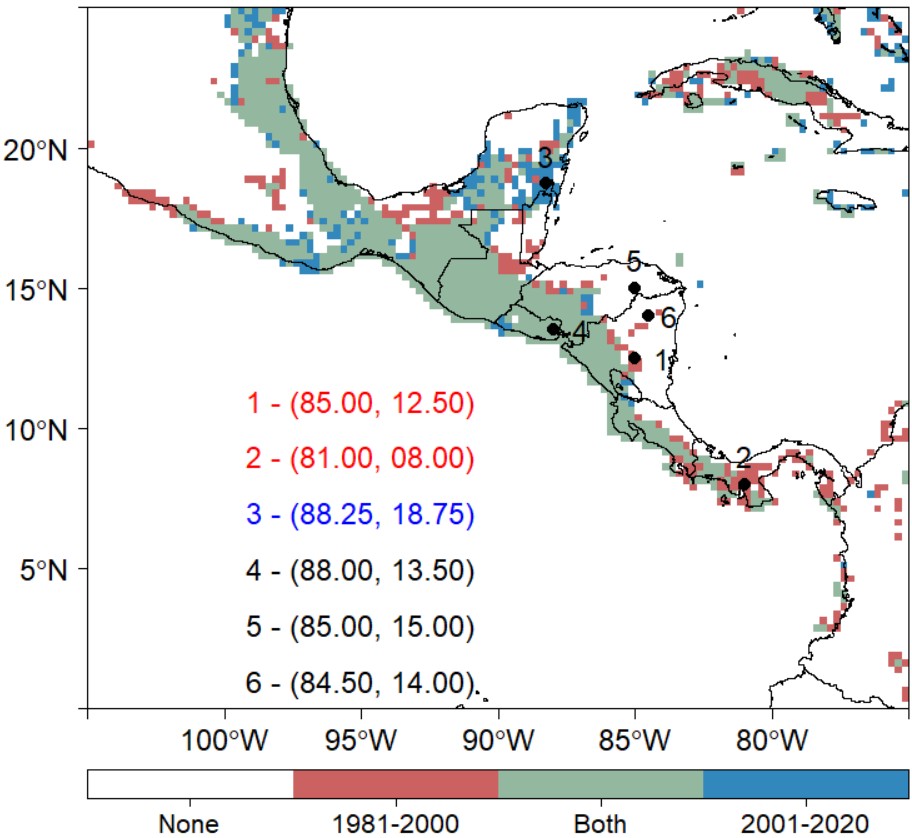

**Figure 4: Occurrence and change in the occurrence of the MSD using the original definition of the MSD (Table 1 and Fig. 1).**
**Shading indicates pixels with an MSD for the early (1981-2000), late (2001 - 2020), or both periods. Also shown are specific points**
**used in subsequent examples or discussion.**

For illustration of the types of details encountered by the classification scheme in any year, Fig. 5 shows for different
locations (identified in Fig. 4) a sample of several time series of precipitation, after applying the smoothing described above.
This also shows the outcome of applying the criteria as to whether an MSD exists in the year depicted.





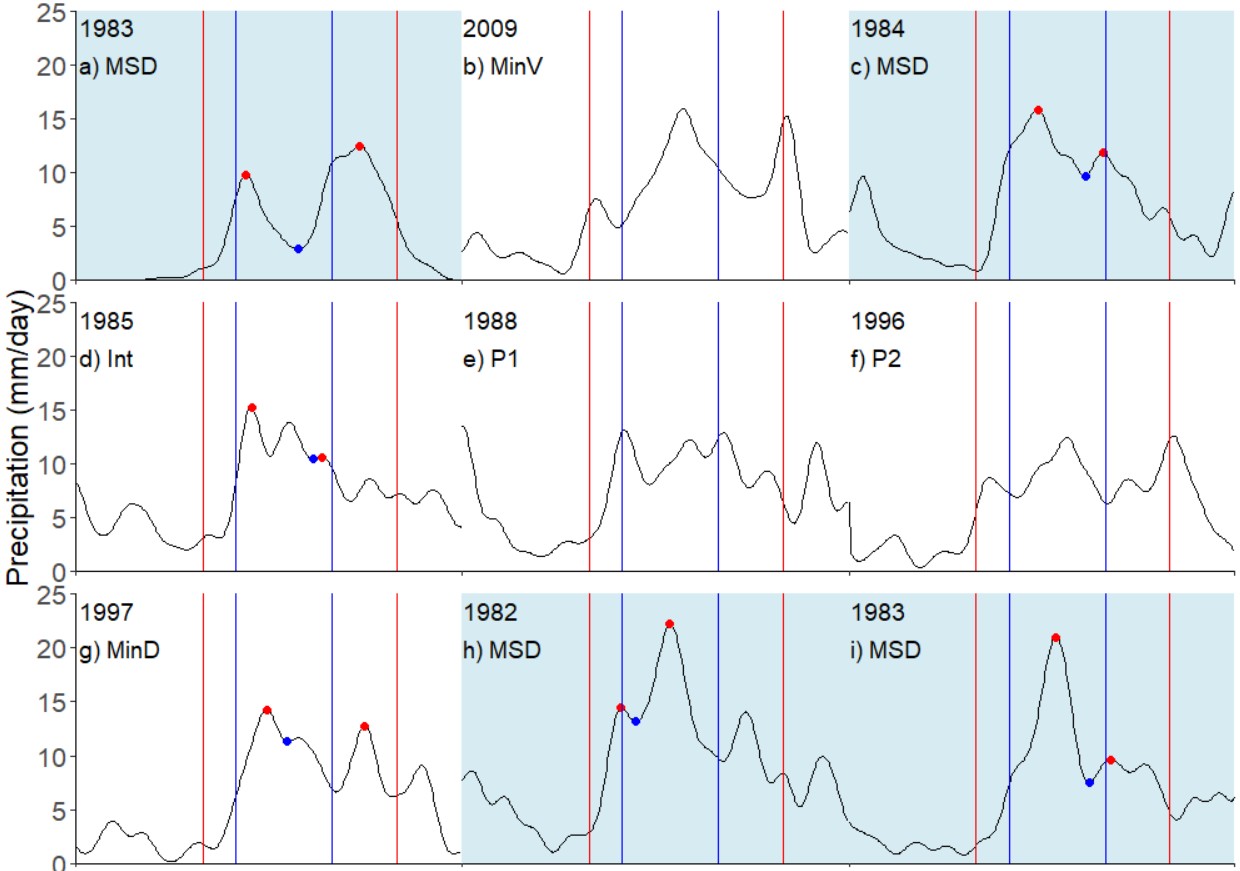

**Figure 5: Daily precipitation time series for the year indicated in the upper left corner of each panel. Blue shading indicates the year is classified as an MSD using our original definition (Table 1 and Fig. 1). Key time windows from Table 1 are indicated by vertical lines: blue lines correspond to window 1 in which a minimum is identified; red lines to window 2 in which peaks must occur. Red dots indicate first and second maxima and blue dots mark the minimum if they meet MSD criteria. Examples shown are a) a canonical MSD pattern at pt. 4, b) no minimum in window 1 at pt. 6, c) high minimum at pt. 5, but still an MSD, d) insufficient intensity at pt. 5, e) early peak is outside window 2 at pt. 6, f) second peak is outside window 2 at pt. 6, g) lower minimum occurs in window 2 at pt. 5, h) and i) high variation in peaks at pt. 5.**

Figure 5 shows that, in most cases, even after smoothing the precipitation signal remains noisy. While the canonical MSD pattern (Fig. 5a) is what is often depicted in the literature (e.g., Anderson et al., 2019; Karnauskas, 2013), MSD years can have a wide variety of shapes in the precipitation record (Fig. 5c, h, i). Years that might appear to be an MSD may fail one or more criteria (Fig. 5d, f, g). These examples highlight the potential sensitivity of MSD classification to the details of its definition. Similar problems can arise when defining the MSD using alternative methods, or even for the date of onset or demise of the rainy season. Several criteria have been developed for these two purposes with advantages and drawbacks (e.g., Alfaro, 2014; Maldonado et al., 2016; Bombardi et al., 2017).

To explore the characteristics affecting the classification of certain locations as having an MSD, we explore three points. Specifically, we examine points that either are classified as having an MSD in the early period (1981-2000) but not in the later period (2001-2020) or vice versa. Figure 6 illustrates the 20-year average conditions at points 1, 2, and 3 (Fig. 4).

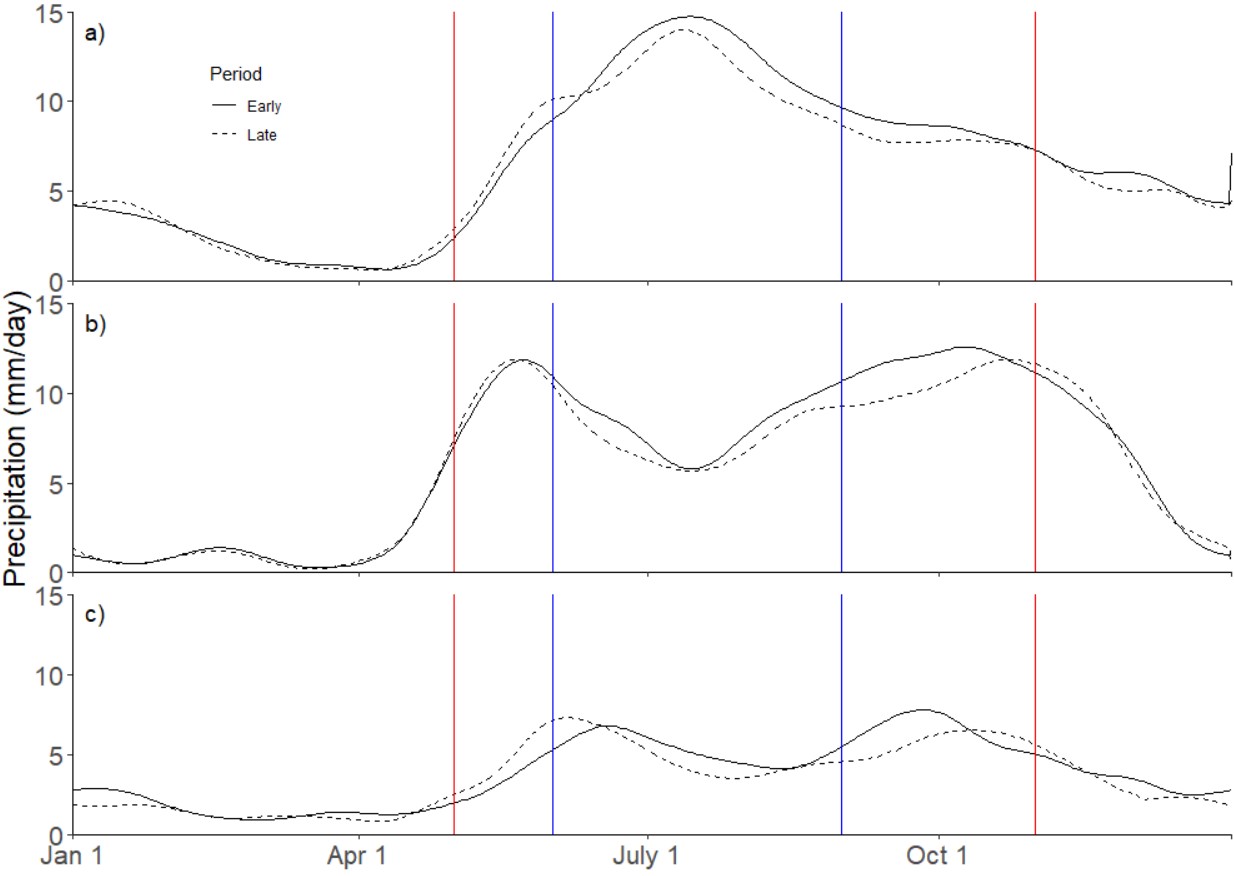


**Figure 6: 20-year average precipitation (in mm/d) pattern at a) pt. 1 with MSD in 1981-2000 but not 2001-2020, b) pt. 2 with MSD in 1981-2000 but not 2001-2020, and c) pt. 3 with MSD in 2001-2020 but not 1981-2000. As with Fig. 5 key time windows are indicated by vertical lines.**

While Fig. 6 shows relatively subtle changes in average precipitation between the 20-year periods at all three points,

underlying these are more systematic changes that affect the MSD classification. Point 1 does not show an MSD signal at all in the average precipitation for either period in Fig. 6a. This is because the typical pattern of precipitation has a single larger peak falling near the center of the 6/1-8/31 window (in which a search is done for a minimum), with the minimum occurring closer to the extreme dates of this window, occurring equally before or after the larger peak, similar to panels h) and i) in Fig. 5. Point 1 shows an overall reduction in precipitation for the later period, especially from June through October. The

declining precipitation produces smaller peaks resulting in a declining intensity (in years classified as an MSD), from 4.8 mm in 1981-2000 to 3.7 mm in 2001-2020, reducing the number of years satisfying the MSD criteria from 16 of 20 years in 1981-2000 to 9 in 2001-2020.





The average precipitation pattern for point 2 shows a more typical MSD pattern for both periods. Similar to point 1, the
1981-2000 period is classified as having an MSD, while 2001-2020 is not. However, the changes are much more subtle, with
the MSD years having nearly the same intensity for both periods (7.9 mm/d and 8.1 mm/d for the early and late periods,
respectively). At point 2, Fig. 6b shows the shift of the second peak to slightly later in the season, which is the important
change at this location. For 1981-2000 18 of 20 years are classified as MSD, while 2021-2020 has 15 years of MSD, falling
just below the threshold of 16 years required for an MSD location. In every case for both periods, the cause of a year not
being an MSD is the second peak slightly outside the 10/1 window required by the definition. Thus, at this location it is the
timing of the second rainfall pulse that changes the MSD classification.

Similar to point 2, point 3 (Fig. 6c) shows an average shift in the second precipitation peak to later in the season for 2001-
2020 compared to 1981-2000 with a slight reduction in the magnitude of the second peak. However, point 3 shows the
opposite effect, with 13 of 20 years classified as MSD for 1981-2000 increasing to 18 for 2001-2020, showing a dramatic
increase in the number of years satisfying the definition of an MSD. Average statistics do not explain the difference, with a
duration of 105 days for 1981-2000 and 107 days for 2001-2020; intensity is relatively constant at 7.4 mm for 1981-2000
and 7.1 mm for 2001-2020. In 1981-2020 the dominant cause for failing to meet the MSD criteria is peak precipitation
occurring outside of the established MSD windows, often in December-January. There is no evident reduction in average
rainfall during December-January in Fig. 6c, nor is there a significant reduction at this location in December-February
rainfall detected in a prior study (Stewart et al., 2021), so the effect is limited to peak events, but it has a strong impact on
MSD classification.

As the values of the different criteria for classifying an MSD vary there are changes in the number of MSD grid cells and
where the MSD occurs. Table 2 provides a summary of how the number of grid cells varies, and the following figures
illustrate changes in the location of the MSD.

Since the precipitation values are smoothed with a 31-day filter, durations shorter than this have no effect on the
classification of MSD grid cells. Thus, while we imposed a minimum 15-day duration as our original definition, changing
this to 30 days would have no effect on results. Imposing a stricter requirement for longer durations reduces the number of
MSD grid cells by about 12% for durations of up to 50 days, but does not substantially change the differences in MSD extent
between the two periods from the original. The general insensitivity of MSD classification to the minimum duration is
consistent with the majority of significant trends in duration being positive (Fig. A5), and focused on grid cells on the Pacific
side that are classified as MSD cells for both periods. Duration definitions of 60 and more days, unsurprisingly reduce the
number of cells exhibiting an MSD pattern by 30% and more. An MSD of that length is also inconsistent with the nature of
the phenomenon as described by smallholder farmers and prior studies.





**Table 2: Columns a), b), and c): total number of MSD grid cells (percent change from original in parentheses) with different values of criteria used to define an MSD for the early period (1981-2000), late period (2001-2020), and both periods. Changes in the total number of MSD pixels between the periods are shown in column d.**

|  | a) Early period (+overlap) | b) Late period (+overlap) | c) Both periods (overlap) | d) % Change between 2 periods |
|---|---|---|---|---|
| Original Definition (Duration=15 days, Intensity=3 mm, 16 out of 20 years (80%)) | | | | |
|  | 1174 | 1075 | 869 | -8.4 |
| Parameter Variations | | | | |
| Duration | | | | |
| 10 Days | 1174 (0%) | 1075 (0%) | 869 (0%) | -8.4 |
| 20 Days | 1174 (0%) | 1075 (0%) | 869 (0%) | -8.4 |
| 30 Days | 1174 (0%) | 1075 (0%) | 869 (0%) | -8.4 |
| 40 Days | 1151 (-2.0%) | 1072 (-0.3%) | 859 (-1.2%) | -6.9 |
| 50 Days | 1054 (-10.2%) | 1003 (-6.7%) | 767 (-11.7%) | -4.8 |
| 60 Days | 841 (-28.4%) | 788 (-26.7%) | 565 (-35.0%) | -6.3 |
| Intensity | | | | |
| 1 mm | 2260 (92.5%) | 2137 (98.8%) | 1953 (124.7%) | -5.4 |
| 2 mm | 1792 (52.6%) | 1568 (45.9%) | 1416 (62.9%) | -12.5 |
| 4 mm | 652 (-44.5%) | 642 (-40.3%) | 481 (-44.6%) | -1.5 |
| 5 mm | 316 (-73.1%) | 360 (-66.5%) | 229 (-73.6%) | 13.9 |
| Window | | | | |
| 2 weeks before | 766 (-34.8%) | 582 (-45.9%) | 447 (-48.6%) | -24.0 |
| 2 weeks after | 1202 (2.4%) | 1155 (7.4%) | 924 (6.3%) | -3.9 |
| % of Years with MSD signal required | | | | |
| 70% (14 of 20) | 1590 (35.4%) | 1466 (36.4%) | 1276 (46.8%) | -7.8 |
| 90% (18 of 20) | 669 (-43.0%) | 610 (-43.3%) | 452 (-48.0%) | -8.8 |






Figure 7 shows the effect of varying the intensity criterion of the original MSD definition (3 mm) between 1 mm and 5 mm. Allowing the low intensity threshold for an MSD classifies nearly the entire domain as having an MSD, with the exceptions being only along the Caribbean side of Central America and most of Colombia. Requiring a more extreme 5 mm intensity for an MSD classification limits zones with MSD to a relatively thin band along the Pacific side of Central America. This larger intensity threshold excludes some areas, such as Northern Nicaragua, where the MSD is a well-known phenomenon, indicating that a higher intensity threshold may not be appropriate.

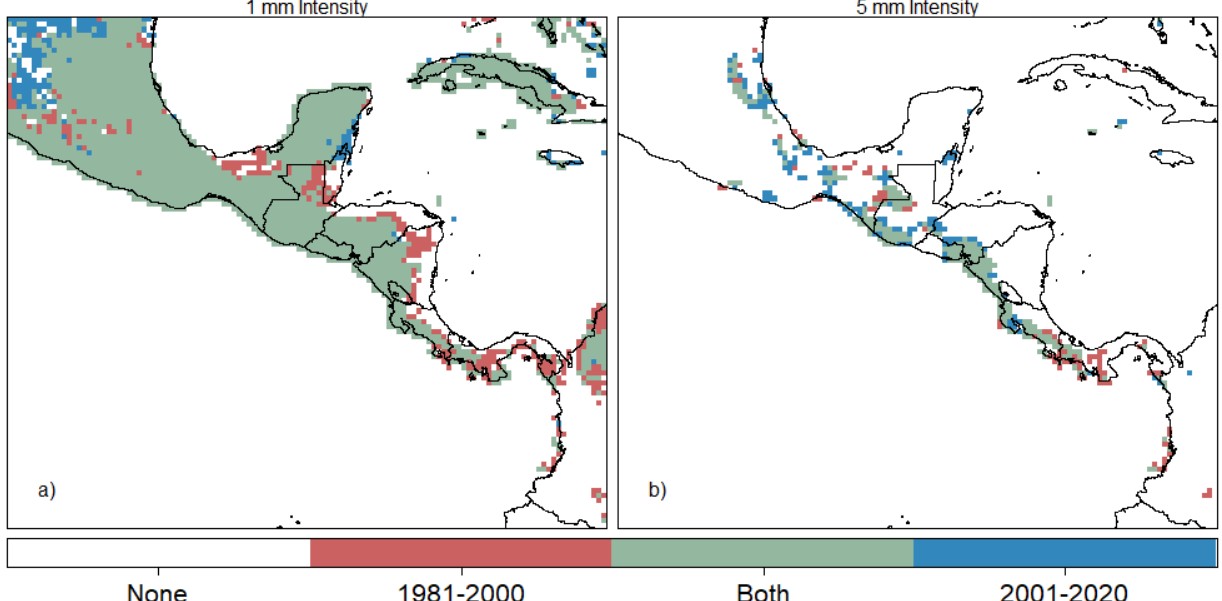

**Figure 7: Pixels showing MSD by varying from the original definition the minimum required intensity from a) 1 mm and b) 5 mm.**

Figure 7 also shows the same spatial pattern of changing MSD grid cells as Fig. 6, with isolated areas in the northern part of the domain changing from not experiencing an MSD to being classified as MSD in the latter half of the study period. Figure 8 highlights the changes between 1981-2000 and 2001-2020 in total MSD grid cells for the domain. Intensity thresholds of less than 4 mm cause many more cells to be included overall, and produces a net reduction in the overall number of cells classified as having an MSD over the last 40 years. The highest threshold isolates only those grid cells that experience the most intense MSD, and also reveals an increase in MSD area. Thus, areas that have historically experienced lower intensity MSD events have contracted in spatial extent over the last four decades. Conversely, areas with historically high intensity MSD events have expanded.





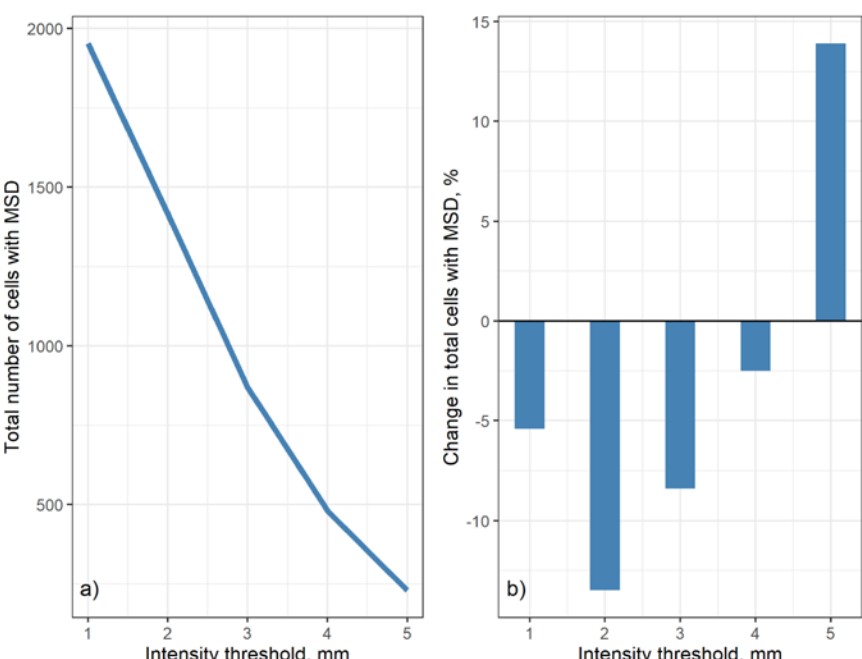

**Figure 8: a) Total number of MSD pixels (only those classified as MSD in both periods), and b) change in the number of pixels classified as experiencing an MSD between 1981-2000 and 2001-2020.**

As was illustrated in Fig. 5, peaks or minima can fall outside of the defined windows by only a day or two and cause a year to not be classified as an MSD. To explore this, we varied the dates of the windows by shifting them all uniformly two weeks earlier and two weeks later. The results are shown in Table 2 and Fig. 9. By shifting the dates earlier (Fig. 9a) there is a dramatic reduction in the area with an MSD, and the later period sees a steeper reduction, increasing the magnitude of the reduction in MSD area between the two periods. Shifting the dates later (Fig. 9b) increases the area classified as having an

MSD for both periods, with three times the increase in area from the early to the later period compared to shifting the dates earlier. This indicates that a more extensive MSD exists later in the season in general, and that there has been a shift in the last 40 years toward a later MSD.





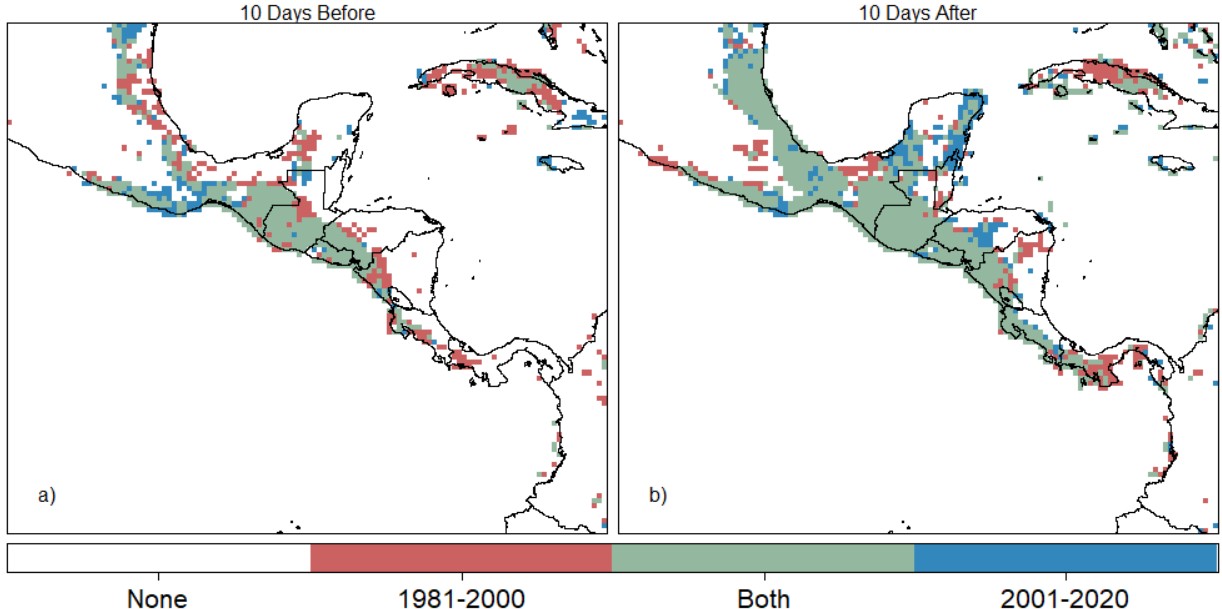

**Figure 9: Pixels with an MSD using the original definition varying the dates associated with the windows for the minimum and**
**peaks from a) two weeks earlier to b) two weeks later.**

Finally, the effect of modifying the number of years any grid cell must have an MSD to be classified as a location with an

MSD is shown in Table 2 and Fig. 10. Adopting a looser (70%) or stricter (90%) requirement for an MSD grid cell changes

the extent, but has little effect on the spatial patterns during each period or the changes between the two periods.

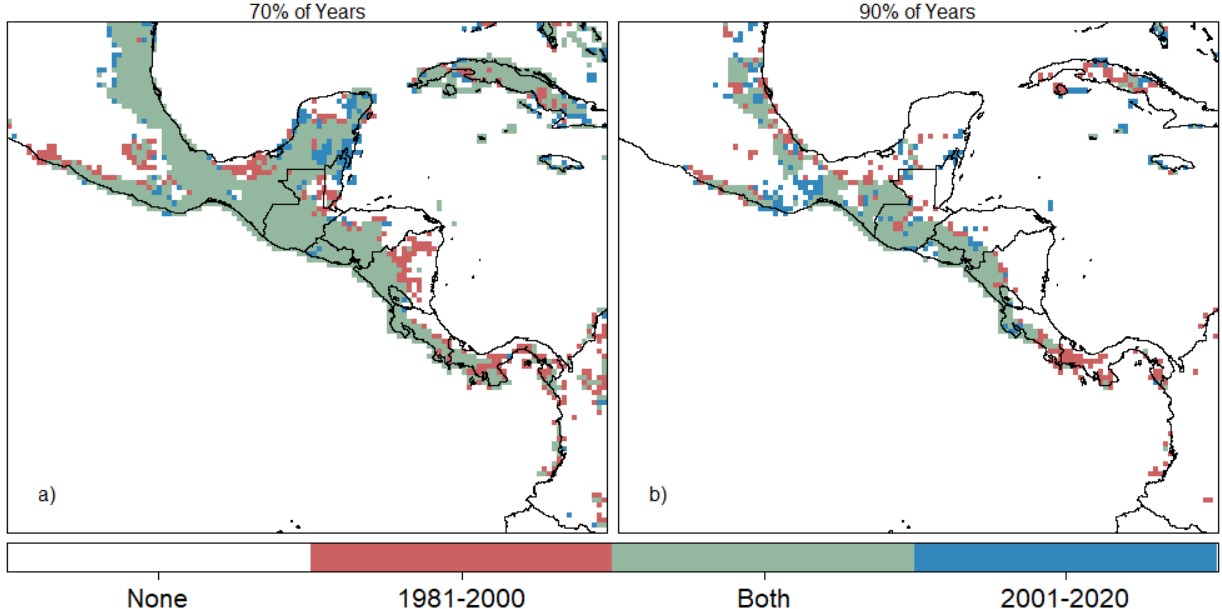

Figure 10: Pixels with an MSD by varying the original definition so that a cell must have an MSD in at least a) 70% of years and b) 90% of years.

## 5 Discussion and Conclusions

The Mesoamerican MSD is typically defined by a set of precipitation characteristics. As studies explore the existence of historical trends or future projections of characteristics of the MSD, understanding the impact of decisions regarding the MSD definition is essential since these can affect results. We found that seasonal variability can cause individual years with detectable MSD signals to be indiscernible in a climatological average, highlighting the importance of assessing individual events for the presence of an MSD.

We examined the four precipitation characteristics defining the dry period between two peaks, centered in July and August: duration (the time between peaks), intensity (the level of decline between the two peaks), the timing (the dates defining windows within which the minimum and peaks must occur), and consistency (the percentage of years with a defined MSD occurring. Of these four, the two having the greatest impact on results were intensity and timing.

The application of a minimum intensity has a dramatic effect on the results of both the area considered as having an MSD and the changes in the recent historical record. Our results suggest that the intensity chosen can affect both the magnitude and direction of changes in the recent observed record. The regions with MSD of greatest intensity show a net increase in area, while areas with a characteristically lower intensity MSD are decreasing in extent. This may reflect the increases in

more extreme precipitation levels in the region, resulting in an intensified MSD, something projected as the climate continues to warm (Maloney et al., 2014; Vichot-Llano et al., 2021).


The original timing established for defining the Mesoamerican MSD definition was that a minimum precipitation should occur in the June 1st - August 31 window, and that a peak inflection should exist on either side of it within the May 1st - October 31 window. Shifting these dates two weeks earlier dramatically reduced the area with an MSD, and shifting them two weeks later increased the area. In addition, shifting the dates in either direction had a strong influence on the observed

change in MSD extent between 1981-2000 and 2001-2020, suggesting a change in precipitation timing to later in the year, so the typical MSD pattern may not be occurring during the time it has historically. MSD timing, and its accurate prediction, is a challenge that could benefit socio-economic sectors throughout the study region (Alfaro et al., 2018). Thus, whether examining past or future changes in MSD or developing improved seasonal forecasts, the non-stationarity of MSD timing should be accommodated.


These results suggest that for studies of historical or future changes in MSD for this region, studies should be conducted for different levels of MSD intensity and timing to capture differing impacts as these characteristics vary across the domain. A greater understanding of the impact of the objective definition of the MSD on changes in the timing, intensity, and the frequency of occurrence of the MSD pattern, and their relative importance to smallholder agriculture, may support

assessments of climate change impacts connected to the MSD and the development of adaptation strategies.

While including temperature effects on MSD determination will become important as effects are projected later in the 21st century, this study includes only precipitation characteristics, since temperature effects are secondary for the recent historical period on which we focus. MSD classification will be affected by ongoing larger scale changes in hydroclimate, for example

with increasing aridity (Karmalkar et al., 2011; Hidalgo et al., 2013) reduced or delayed seasonal peak precipitation could lead to a year being a non-MSD year, despite prevailing intense drought conditions.

**Author contribution**

EM, IS: Conceptualization; data curation; formal analysis; funding acquisition; investigation; methodology; project administration; visualization; writing - original draft; writing-review & editing. KJ: Investigation, software, visualization.

HH: Formal analysis, writing-review & editing.





**Competing interests**

The authors declare that they have no conflict of interest.

**Acknowledgements**

We gratefully acknowledge funding for this work from U.S. National Science Foundation Grant BCS-1539795, DFG (German Research Foundation) Grant STA 632/6-1, and the Freiburg Institute of Advanced Studies (FRIAS) at Freiburg University (Germany). Work by K. Joseph was in part supported by the Environmental Justice and the Common Good Initiative at Santa Clara University. Our understanding of the midsummer drought greatly benefited from discussions with director Raúl Díaz and coworkers at CII-ASDENIC and smallholder farmers in northern Nicaragua. Our work has also

benefited from details on their work shared by Anderson et al. (2019). We thank Sven Decker for his work on Fig. 2.

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





**Appendix A**

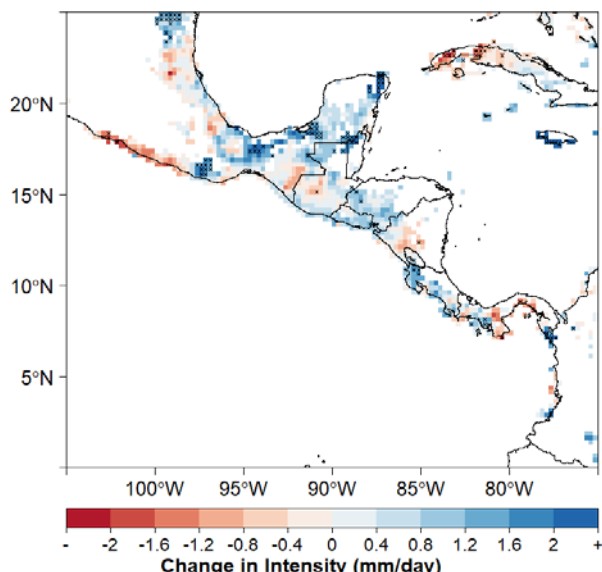

**Figure A1: The change in MSD intensity between the early (1981-2000) and late (2001-2020) periods. Grid cells marked with an**
**"X" show statistically significant changes.**

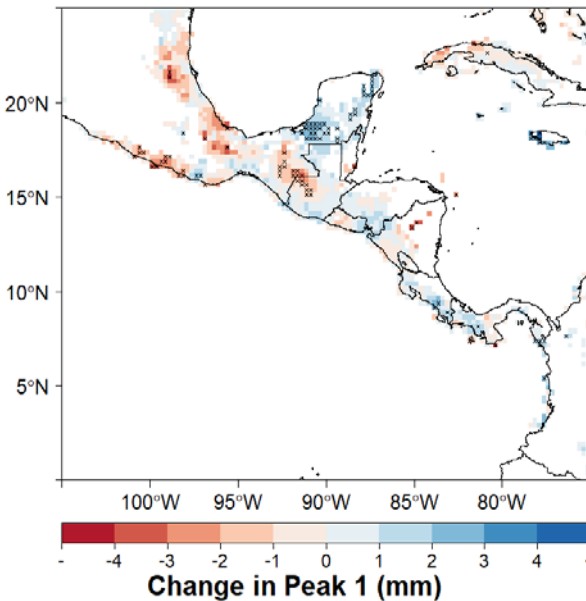

**Figure A2: The change in magnitude of the first peak MSD between the early (1981-2000) and late (2001-2020) periods. Grid cells marked with an "X" show statistically significant changes.**



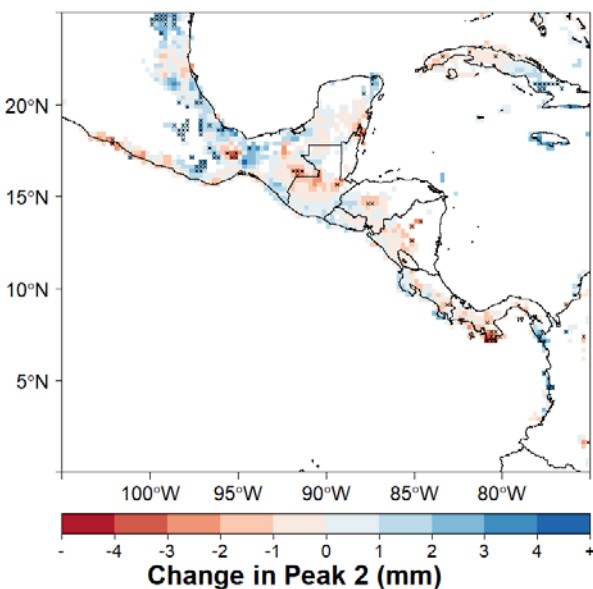

**Figure A3: The change in magnitude of the second peak MSD between the early (1981-2000) and late (2001-2020) periods. Grid cells marked with an "X" show statistically significant changes.**

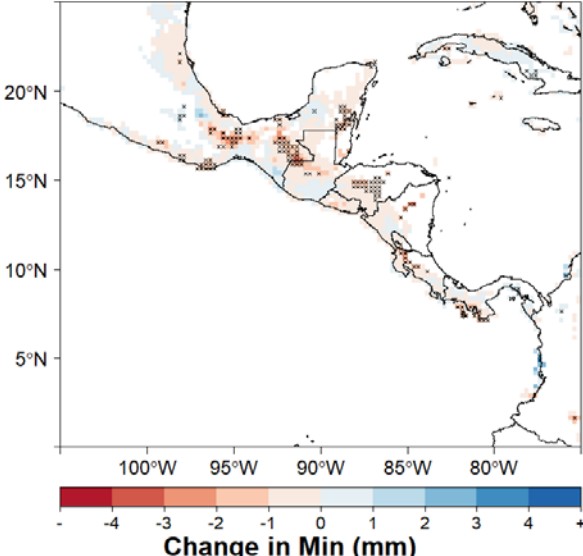

**Figure A4: The change in MSD minimum between the early (1981-2000) and late (2001-2020) periods. Grid cells marked with an "X" show statistically significant changes.**





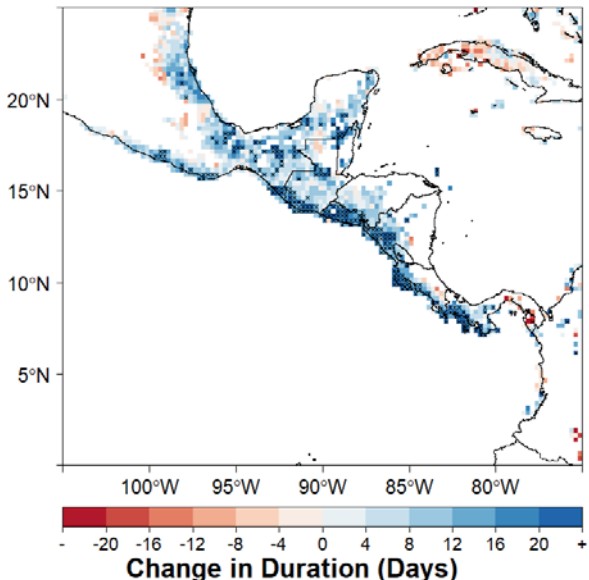


**Figure A5: The change in MSD duration between the early (1981-2000) and late (2001-2020) periods. Grid cells marked with an "X" show statistically significant changes.**