# Peer review of "The Mesoamerican mid-summer drought: the impact of its definition on occurrences and recent changes"

_Hydrology and Earth System Sciences, 2021_

## Author Comment (AC1)

Final response to reviewer comments on HESS 2021-543, The Mesoamerican mid-summer drought: the impact of its definition on occurrences and recent changes.

We are grateful to the reviewers for their careful review and thoughtful comments on the submitted manuscript. Our detailed responses to each comment follow. Reviewer comments are in regular type. Responses are indented and in red.

RC1: 'Comment on hess-2021-543', Anonymous Referee #1

I would like to recommend the paper to be accepted at HESS, but just after the improvement of some figures and points.

Minor comments:

1. Please improve the quality of figures 2 and 3.

> Figures 2 and 3 have been re-created in a higher quality format for the revised manuscript. In addition, Figure 2 has been rearranged with the panels stacked vertically to allow each to be somewhat larger.

2. Line 122, page 4, change: (1981 - 2000) to (1981-2000)

> This change has been made in the revised document.

3. Line 191, page 9, change: late (2001 - 2020) to late (2001-2020)

> This change has been made in the revised document.

4. Line 311, page 17, a parenthesis is missing "(the percentage of years with a defined MSD occurring."

> This change has been made in the revised document.

RC2: 'Comment on hess-2021-543', Anonymous Referee #2

The authors present a manuscript that evaluates how varying the definition of the Central American mid-summer drought (MSD) impacts the magnitude of observed changes over the last 40 years. The results are of interest to the climate science community because they find that adjusting certain parameters in the MSD definition has significant impacts on changes across the observational record. The manuscript includes a new definition of the MSD in an expanded region, the use of daily data, and analyses of how the MSD parameters influence the results. It is important, as they show that the potential non-stationarity of the MSD should be accounted for in all future analyses. I recommend that this manuscript is accepted at HESS after addressing the following minor revisions.

Line 32: "Many MSD descriptions have originated with the smallholder farmers in the region and often have qualitative components." This sentence seems out of place since no examples of qualitative definitions are provided in this paragraph. I recommend the authors either provide some qualitative definitions to show how they vary or remove this sentence since it appears to be unrelated.

> While we have some other documented definitions from field surveys, we agree that simply striking this sentence makes the paragraph concise and clear. We removed that sentence.

Line 42: Add "observational" between "recent record".

> This change has been made in the revised document.

Line 73: Please clarify what you mean by "most recent warming".

> This sentence has been rewritten to address this ambiguity: ""However, we are not aware of a study that has examined the sensitivity of the MSD spatial and temporal extent to its definition, and the impact the definition has on the presence of Central America wide changes during the warming trends over the past four decades on the order of 0.8°C·decade−1 (Stewart et al., 2021).

Line 89: I understand that the CHIRPS data was aggregated in the previous study referenced in this sentence (Stewart et al. 2021) for comparison against other precipitation datasets. Please provide a reason for why the data was aggregated in this study since no comparisons against other precipitation datasets are made. I also ask that the authors address how this resampling may affect the characterizations of the MSD.

> Thank you for pointing out this shortcoming. We have added to that paragraph the following "The data were aggregated to reduce data volumes and facilitate exploration of the influence of different MSD definitions. To verify that this aggregation does not affect the results of this analysis, Fig. A1 and Table A1 show results for a reduced area in Central America using data at both the original CHIRPS resolution and the aggregated resolution, with consistent results at both scales."

> The referenced Figure A1 and Table A1 are added to the Appendix, consisting of the following:

[Figure]

**Figure A1: Number of years out of 20 with an MSD in the early and late periods using 0.25° aggregated data (a, b) and original CHIRPS resolution of 0.05° (c, d). Circles in panel b indicate the difference in proportions of MSD years in the two periods is statistically significant based on Fisher's exact test.**

Table A1: Summary of percent of grid cells with MSD and the change between the early and late periods, based on the original MSD definition.

| Resolution | % of Cells with MSD - Early | % of Cells with MSD - Late | Change in % of Cells | % of cells with significant differences |
|---|---|---|---|---|
| 0.25° | 61.4 | 54.7 | 6.7 | 2.0 |
| 0.05° | 56.7 | 50.8 | 5.9 | 2.3 |

Line 94: Please reword this sentence as it is confusing as written. Maybe switch to something like "We use the CHIRTS dataset for the limited temperature analysis in this paper".

> This sentence has been revised as recommended by the reviewer.

Line 119: I recommend that the authors explicitly state the values for the parameters they change in the MSD definition in the methods (i.e. durations, intensities, windows, % of years with an MSD required). For example, at line 119 list the different thresholds that are analyzed for % of years with an MSD required.

> We deleted the phrase "..., and we explore the effect of changing this value" at that location, since it was redundant with later text. To address this comment we modified Table 1 to include both the original values and the range over which each variable is varied, as shown below.
>
> Table 1: Variable values for key components of the MSD definition.

| Variable | Original Values | Range of Values Explored |
|---|---|---|
| Minimum duration | 15 days | 10 – 60 days |
| Minimum intensity | 3 mm/day | 1 – 5 mm/day |
| Window 1 | June 1st - August 31 | 2 weeks earlier, 2 weeks later |
| Window 2 | May 1st - October 31 | 2 weeks earlier, 2 weeks later |
| % of years | 80% (32 of 40 years or 16 of 20 years) | 70 – 90 % |

Line 120: I think it is important that the authors address how changing the two main periods of analysis could influence the results? Since you change many other factors, it might be relevant to consider moving windows (i.e. 10 years) for the time periods.

> The time periods being compared would absolutely influence the results of the changes reported in this work. The time periods compared are not a part of the MSD definition, and thus were not varied. The use of the 20-year periods is explained more clearly in the revised paragraph:

"To explore how the definition of the MSD affects the changes in MSD over the recent past, we divide the study period into two equal 20-year periods, namely (1981-2000) and (2001-2020) and compare average MSD conditions between the earlier and the later period. Especially since these periods are shorter than the recommended 30-year length for establishing a climate normal (WMO, 2017) they should not be interpreted as definitely quantifying changes in the MSD. Rather, these are used principally to illustrate the influence variations in MSD definitions have on the detection of shifts in MSD characteristics or location."

Figure 2: Please include a higher resolution version of this figure. It is really challenging to see the patterns and differences for individual grid cells.

Figures 2 has been re-created in a higher quality format for the revised manuscript. See the response to RC1 comment 1.

Line 155-156: Please provide clarification since parts of this sentence seem to contradict each other. It says temperature change is highly variably month to month, but that they are also mainly positive and statistically significant.

To address this lack of clarity, we have revised the sentence to the following: "Warming on the order of 1-2 °C has generally taken place throughout the domain. Observed temperature increases are variable month-to-month, though changes are broadly positive and statistically significant both seasonally and annually. Less significant warming is observed in September-November (Fig 2b; Stewart et al., 2021)."

Figure 3: Clarify by adding "Number of years out of 20 with a MSD"

This change has been made in the revised document.

Line 167: Please clarify what variable is being referenced in terms of statistically significant changes (i.e. statistically significant changes in the number of years with a MSD)

We have clarified this in the revised text with the suggested language: "In Fig. 3, two-thirds of the grid cells with statistically significant changes in the number of years with an MSD are locations that are dominated by an MSD in the early period but not in the late period"

Figure 4: Please address how the individual points were selected in the methods.

The caption for Figure 4 has been changed to include "Also shown are specific points used in subsequent examples or discussion, selected to show a variety of MSD characteristics and different changes between the early and late periods."

Figure 6: I recommend the authors mark the day 1 of every month instead of just Jan, Apr, Jul, Oct.

This change has been made in the revised document.

Line 274: Are you meaning to compare the spatial patterns in figure 7 with figure 6? It seems like you might actually mean to reference Figure 4?

Yes, thank you for catching that error. The revised text references Figure 4.

Lines 276-279: Please clarify the logic of this sentence since it is confusing as written. Maybe instead write something like "…net reduction in the overall number of cells classified as having an MSD when considering the difference between the two periods".

That sentence has been reworded to "An intensity threshold of less than 4 mm causes many more cells to be included overall and produces a net reduction between the two periods in the number of cells classified as having an MSD."

Figure 9: I recommend the authors make this a 3-panel figure that includes the original definition for direct comparison with the different thresholds.

We appreciate this comment, but after some consideration we prefer to leave this figure as is. The results using the original definition are shown in Figure 4. For consistency, if we were to include a center panel with the original definition in Figure 9, we would also need to include the same additional panel in Figures 7 and 10. That would result in redundancy with Figure 4 appearing four times in a relatively short paper. Also, an additional panel would cause the reduction in size of the figure, which could make observing the details more difficult. That being said, we will defer to the opinion of the editorial staff if a preference is to add the panel to all affected panels.